# *OsASR6* Enhances Salt Stress Tolerance in Rice

**DOI:** 10.3390/ijms23169340

**Published:** 2022-08-19

**Authors:** Qin Zhang, Yuqing Liu, Yingli Jiang, Aiqi Li, Beijiu Cheng, Jiandong Wu

**Affiliations:** National Engineering Laboratory of Crop Stress Resistance Breeding, School of Life Sciences, Anhui Agricultural University, Hefei 230036, China

**Keywords:** salt tolerance, *Oryza sativa*, OsASR6, OsNCED1, abscisic acid

## Abstract

High salinity seriously affects crop growth and yield. Abscisic acid-, stress-, and ripening-induced (ASR) proteins play an important role in plant responses to multiple abiotic stresses. In this study, we identified a new salt-induced *ASR* gene in rice (*OsASR6*) and functionally characterized its role in mediating salt tolerance. Transcript levels of *OsASR6* were upregulated under salinity stress, H_2_O_2_ and abscisic acid (ABA) treatments. Nuclear and cytoplasmic localization of the OsASR6 protein were confirmed. Meanwhile, a transactivation activity assay in yeast demonstrated no self-activation ability. Furthermore, transgenic rice plants overexpressing *OsASR6* showed enhanced salt and oxidative stress tolerance as a result of reductions in H_2_O_2_, malondialdehyde (MDA), Na/K and relative electrolyte leakage. In contrast, *OsASR6* RNAi transgenic lines showed opposite results. A higher ABA content was also measured in the *OsASR6* overexpressing lines compared with the control. Moreover, OsNCED1, a key enzyme of ABA biosynthesis, was found to interact with OsASR6. Collectively, these results suggest that OsASR6 serves primarily as a functional protein, enhancing tolerance to salt stress, representing a candidate gene for genetic manipulation of new salinity-resistant lines in rice.

## 1. Introduction

Salinity is a major abiotic stress factor, seriously affecting crop growth and yield, and causing substantial economic losses as well as threatening global food security. Studies suggest that more than 7% of the global land area (~800 million hectares) will be affected by salinity [1]. Salt accumulation in the soil, mainly in the form of sodium chloride (NaCl) from irrigation water [2] leads to osmotic and ionic stress [3]. Osmotic stress affects water absorption and stomatal closure, inhibiting plant growth [1], while ionic stress causes ion toxicity, impairing photosynthetic machinery [4]. In addition, both osmotic and ionic stress result in the production of excessive reactive oxygen species (ROS), which leads to oxidative damage in plants [5].

To combat the harmful effects of salt stress, plants have evolved a number of regulatory strategies, including physiological, biochemical and morphological responses. For example, plants are able to reduce the ratio of Na/K by reducing Na^+^ ion uptake and removing excess Na^+^ from cells facing salt stress [6]. Plants are also able to enhance salt tolerance via abscisic acid (ABA) and H_2_O_2_ signaling. ABA is an important phytohormone that plays an important role in regulating tolerance to a number of abiotic stresses. Stress-responsive genes can be divided into two types: ABA-dependent and ABA-independent, depending on their involvement in stress perception and signaling. Under salt stress, expression of ABA synthesis related or ABA-inducible genes is activated in order to protect against damage. For example, the application of exogenous ABA and increased levels of endogenous ABA via overexpression of stress-response genes were found to reduce damage under various types of stresses, respectively [7].

Of these regulatory mechanisms, transcription factors (TFs), by binding target genes, and molecular chaperones, by interacting with key stress-related proteins, are crucial in enhancing stress tolerance. Increasing research suggests that TFs play important roles in salt stress responses by controlling expression of target genes [8]. For example, AP2-type transcription factor *OsSAE1* was found to positively regulate tolerance to salt by directly binding to the *OsABI5* promoter, repressing ABA signaling in rice [9]. Moreover, increased protein stability and prevention of protein aggregation by molecular chaperones were found to effectively increase tolerance to abiotic stress [10].

ABA-, stress-, and ripening-induced (ASR) proteins are a small family of proteins widely found in most plant species, all of which contain a conserved ABA/water deficit stress (WDS) domain in the C-terminal region [11]. *ASR* genes have been shown to regulate plant senescence, fruit development and ripening [12], while research also suggests a role in abiotic stress tolerance, and subsequent improvements in crop yield under stress conditions. In line with this, *TaASR1-D* overexpressing wheat was found to show enhanced tolerance to drought and salt through ROS and ABA signaling, with subsequent improvements in productivity [13], while transgenic maize plants overexpressing *ZmASR1* were found to confer drought tolerance, with no effect on yield [14]. In addition, overexpression of *OsASR1* was found to improve rice growth and subsequent yield under drought conditions by regulating stomatal closure [15].

Based on the above results, a number of studies have focused on *ASR* genes in order to combat abiotic stresses. *ASR* genes are thought to serve as TFs, regulating the expression of target genes, or molecular chaperones, protecting reporter enzymes from denaturation [16,17,18,19]. However, the underlying mechanisms remain unknown.

Growth and grain yield in rice, one of the most important staple crops in the world, are seriously affected by salt stress. Identification of new tolerant genes and exploration of the underlying molecular mechanisms are therefore important. Rice contains six *ASR* genes located on four different chromosomes [20]. However, of these, only three (*OsASR1*, *OsASR3* and *OsASR5*) have so far been found to protect plants against abiotic stress [15,17,21]. In this study, we identified a new salt-induced *ASR* gene, *OsASR6,* and characterized its roles in mediating salt tolerance in rice. Transgenic plants overexpressing *OsASR6* exhibited increased tolerance to salt stress, while *OsASR6* RNAi transgenic plants displayed increased sensitivity. In addition, OsASR6 was found to physically interact both in vitro and in vivo with *Oryza sativa* 9-cis epoxycarotenoid dioxygenase 1 (OsNCED1), a key enzyme of ABA biosynthesis. Furthermore, *OsASR6* was found to regulate ROS accumulation and ABA synthesis under salt stress, suggesting that *OsASR6* enhances salt tolerance via H_2_O_2_ and ABA signaling.

## 2. Results

### 2.1. Induced Expression of OsASR6 under Stress Treatment

Analysis of induced expression patterns can effectively help determine the roles of certain genes in response to abiotic stress. Transcription levels of *OsASR6* were therefore analyzed in root and shoot samples obtained under high salinity, H_2_O_2_ and ABA treatment. Under salt stress conditions, *OsASR6* gene expression increased rapidly, peaking after 1 h in the shoots and 3 h in the roots (Figure 1). Moreover, after 1 or 3 h treatment with H_2_O_2_ and ABA, mRNA expression levels increased rapidly in both the roots and shoots (Figure 1). These results suggest that *OsASR6* can be induced by multiple treatments.

### 2.2. Sub-Cellular Localization and Transcriptional Activation Activity of OsASR6

To investigate the sub-cellular localization of OsASR6, we constructed a pCAMBIA1305-OsASR6-GFP vector then transfected the confirmed OsASR6-GFP plasmid into tobacco and rice leaf protoplasts. GFP signals of OsASR6-GFP were subsequently localized in both the nucleus and cytoplasm of the transformed rice protoplasts (Figure 2A). Similar localization of OsASR6-GFP was also observed in tobacco leaf epidermal cells (Figure 2B).

*ASR* genes reportedly serve as either TFs or molecular chaperones [18,19]. In order to clarify this, we analyzed transcriptional activation activity of OsASR6 using a pGBKT7-OsASR6 construct transformed into yeast strain Y_2_HGold. All constructs (pGBKT7-OsASR6, pGBKT7-53 + pGADT7-T and pGBKT7-lam + pGADT7-T) grew well on the SD/-Trp plates; however, the pGBKT7-OsASR6 construct showed no growth on SD/-Trp/-His/-Ade medium (Figure 2C). These findings suggest that OsASR6 has no transcriptional activation activity in yeast.

### 2.3. OsASR6 Positively Enhances Tolerance to Salt in Transgenic Rice

To determine the physiological functions of *OsASR6*, two independent OsASR6-overexpressing (OsASR6-OE, OE19 and OE30) and silenced (OsASR6-RNAi, Ri5 and Ri11) transgenic rice lines were generated then expression levels were examined using qRT-PCR and RT-PCR methods (Appendix A). Germination rates of the WT and transgenic plants were first analyzed under NaCl treatment to confirm the roles of *OsASR6* in salt tolerance. Accordingly, the OsASR6-OE lines exhibited a higher germination rate compared to the WT following 100 mM NaCl treatment for 4 d, while the OsASR6-RNAi lines showed a reduced germination rate (Figure 3A,B). Next, phenotypes of two-week-old WT, OsASR6-OE and OsASR6-RNAi seedlings grown in liquid Hoagland solution containing 130 mM NaCl were observed. No phenotypic differences were observed under normal growth conditions (Figure 3C); however, after NaCl treatment for 14 d, a higher survival rate and lower REL were detected in the OsASR6-OE plants relative to the WT. In contrast, opposite results were observed in the OsASR6-RNAi plants (Figure 3D,E). Moreover, analysis of Na^+^ and K^+^ contents in the shoots revealed a lower Na^+^ content and Na/K ratio in the OsASR6-OE compared to WT plants following 14 d salt stress, in contrast to the OsASR6-RNAi plants, which displayed a higher Na^+^ content and Na/K ratio (Figure 4A–C). Meanwhile, compared with the WT, K^+^ content increased in the OsASR6-OE plants and slightly decreased in the OsASR6-RNAi plants following salt treatment (Figure 4B). These findings suggest that *OsASR6* enhances rice tolerance to salt stress.

### 2.4. OsASR6 Enhances Oxidative Stress Resistance in Rice

*OsASR6* gene expression was rapidly induced following H_2_O_2_ treatment, suggesting that *OsASR6* is also involved in oxidative stress responses. To confirm this, the H_2_O_2_ content of transgenic and WT plants was determined. After treatment with 130 mM NaCl for 14 d, the OsASR6-OE lines displayed a lower H_2_O_2_ content, while the OsASR6-RNAi plants had a higher level compared with the WT (Figure 5A). Meanwhile, DAB staining was then carried out to confirm the H_2_O_2_ levels. Accordingly, leaf samples from the OsASR6-OE plants exhibited a weaker staining intensity, while deep broad staining was detected in the OsASR6-RNAi samples (Figure 5B). ROS detoxifying proteins such as SOD, POD and GPx can effectively decrease the accumulation of ROS. Therefore, physiological indices (MDA contents, and SOD, POD and GPx activities) were also measured to confirm the roles of *OsASR6* in the response to salt stress. Results showed a decrease in MDA, and increases in SOD, POD and GPx activities in the OsASR6-OE lines, in contrast to the OsASR6-RNAi lines, which displayed an opposite effect (Figure 5C–F). Interestingly, no significant differences in APx or CAT activity were obsereved between the transgenic lines and WT under salt stress (Appendix A). These findings suggest that OsASR6 modulates H_2_O_2_ homeostasis in rice by regulating H_2_O_2_-scavenging enzyme activity under salt conditions.

To further identify the function of *OsASR6* during oxidative stress, the role of MV was examined. MV can transfer electrons from the photosynthetic electron transport chain to oxygen, thus inducing the formation of ROS and the creation of oxidative stress conditions in plants. Growth performance following treatment with 2 µM MV for 6 d was also examined. Accordingly, no significant differences in growth or shoot length were observed between the transgenic and WT plants under normal growth conditions (Figure 6A,B). However, following MV treatment, a longer shoot length and increase in chlorophyll were observed in the OsASR6-OE lines relative to the WT. In contrast, the OsASR6-RNAi lines displayed a shorter shoot length and lower chlorophyll content (Figure 6A–C). Collectively, these results suggest that *OsASR6* enhances oxidative stress tolerance in rice.

### 2.5. OsASR6 Regulates ABA Synthesis by Interacting with OsNCED1

It is widely known that ABA is involved in plant responses to salt stress [9,22]. To determine whether *OsASR6* regulates ABA signaling, the sensitivity of WT and transgenic lines were analyzed after 14 d treatment with ABA. Accordingly, the OsASR6-OE lines showed a significant reduction in shoot length compared with the WT, while the OsASR6-RNAi lines exhibited a longer shoot length (Figure 7A–C). These findings suggest that *OsASR6* increases ABA sensitivity in rice.

To determine the possible regulatory mechanism of *OsASR6* in ABA signaling, yeast two-hybrid screening was carried out. To do so, full-length *OsASR6* was inserted into the pGBKT7 vector as bait, revealing several positive clones corresponding to OsNCED1. To confirm this interaction, OsNCED1 was cloned and fused into the pGADT7 vector then OsASR6 and OsNCED1 were co-transformed into yeast. Yeast cells containing either OsASR6 or OsNCED1 were unable to grow on the QDO medium; however, transformed cells containing OsASR6 and OsNCED1 grew well (Figure 8A). Meanwhile, to confirm the interaction between OsASR6 and OsNCED1 in vivo, a BiFC assay was performed. As expected, YFP was detected only following co-transfection with OsASR6 and OsNCED1 (Figure 8B). These results collectively confirm that OsASR6 and OsNCED1 interact both in vivo and in vitro.

NCED1 encodes a key enzyme of ABA biosynthesis [23]. To determine whether the interaction between OsASR6 and OsNCED1 affects ABA synthesis in rice, ABA contents were also determined. Under normal growth conditions, no significant differences in ABA contents were detected; however, following NaCl treatment, an increase in ABA was observed in the OsASR6-OE lines compared with the WT; however, there were no differences in ABA content between the Ri5 and WT plants (Figure 9). These results suggest that OsASR6 regulates ABA synthesis by interacting with OsNCED1 in response to salt stress response.

## 3. Discussion

Since the identification of the *ASR1* gene in tomato [24], increasing research has suggested a role of ASR genes in abiotic stress tolerance in transgenic plants [13,19]. Rice chromosomes possess six paralogous *ASR* genes, and while *OsASR1*, *OsASR3* and *OsASR5* have been shown to enhance abiotic stress tolerance in transgenic plants [15,17,20], little is known about the role of *OsASR6*. In fact, transcription of *OsASR6* (previously named *OsASR4*) was found to be unaffected by drought treatment [17], as confirmed in our previous findings (data not shown). In this study, we therefore characterized the functioning of *OsASR6* under salt stress. The transcription of *OsASR6* was strongly induced by ABA, H_2_O_2_ and high salinity, suggesting a potential role in abiotic stress resistance (Figure 1). Consistent with this, *OsASR6* overexpressing lines showed improved germination and survival rates under high salinity treatment, in contrast to the OsASR6-RNAi lines, which displayed opposite effects (Figure 3A,C). More importantly, *OsASR6* had no effect on growth and development in the transgenic lines. Collectively, these findings suggest that *OsASR6* is a candidate gene for genetic manipulation of new salinity-resistant lines in rice.

Ionic imbalance, or ionic toxicity, primarily occurs as a result of high Na^+^ levels in the cytosol [25]. Meanwhile, increasing research suggests that efficient maintenance of an appropriate Na/K ratio is important for mitigating ionic toxicity during salt stress conditions [26]. In line with this, transgenic plants overexpressing *FcWRKY40*, a WRKY transcription factor, were found to confer salt tolerance by maintaining cellular ion balance [27]. Meanwhile, *Arabidopsis* plants overexpressing *MsPIP2;2*, an aquaporin protein, showed increased salt tolerance due to the improved ratio of Na/K compared to control plants [28]. Meanwhile, maize *wrky86* mutant plants were found to show enhanced tolerance to salt stress as a result of reductions in Na^+^ [29]. Similarly, in this study, the *OsASR6* overexpressing lines presented a reduced ratio of Na/K through reductions in Na^+^ and increases in K^+^ under salt stress conditions, while the OsASR6-RNAi plants showed an increased Na/K ratio (Figure 4). These results suggest that *OsASR6* maintains and reestablishes cellular ion balance, thus enhancing salt stress tolerance in rice.

Excessive ROS accumulation leads to cell membrane damage. Plants have therefore evolved a number of mechanisms to protect themselves against oxidative stress, such as the formation of ROS detoxifying proteins [30]. For example, transgenic wheat overexpressing TaPRX-2A exhibited increased tolerance to salt by regulating accumulation of ROS [31]. Meanwhile, *OsASR5* was found to enhance drought tolerance in rice via H_2_O_2_ signaling [17]. Similarly, the results of this study suggest that *OsASR6* confers tolerance to oxidative stress via physiological responses [17,30,31]. Compared with the WT, OsASR6-overexpressing plants presented a lower MDA content and REL rates, and higher activities of SOD, POD and GPx under NaCl treatment (Figure 5C–F), suggesting that membrane lipids in the transgenic plants were subjected to a lower level of oxidative damage. In addition, reductions in H_2_O_2_ and ROS accumulation were also observed in the OsASR6-overexpressing plants (Figure 5A,B). This is consistent with a previous report in which SiASR4-overexpressing transgenic plants were found to possess increased tolerance to salt due to reductions in ROS accumulation and H_2_O_2_ levels [32]. Moreover, in this study, the OsASR6-overexpressing plants also showed a greater shoot length under MV stress, further confirming a role of *OsASR6* in conferring oxidative stress tolerance. Overall, these results suggest that *OsASR6* enhances salt tolerance by regulating H_2_O_2_ signaling.

ASRs are thought to function as either molecular chaperones or TFs [11]. To confirm the molecular mechanism underlying the response of *OsASR6* to salt stress, we therefore determined the sub-cellular localization of OsASR6 and any transcriptional activation activity. As a result, nuclear and cytoplasmic localization were confirmed (Figure 2A,B), while transcriptional activation activity was absent (Figure 2C), suggesting that *OsASR6* is not a TF. However, interaction between the NCED1 and OsASR6 proteins was identified in chloroplasts of the rice protoplasts, suggest that OsASR6 translocates from the cytoplasm to chloroplasts as molecular chaperones. Previous research suggests interaction of OsASR5 with HSP40, OsASR5 and 2OG-Fe (II) in the protoplasts of rice and tobacco epidermal cells [17]. These results also suggest that *ASRs* in different cellular compartments possess different functional mechanisms under abiotic stress conditions [16]. Furthermore, a higher ABA content was also measured in OsASR6-overexpressing plants under salt stress, suggesting that OsASR6 improves salt tolerance by regulating ABA synthesis. Rice possesses six *ASR* paralogous genes, some of which, including OsASR5, have been found to affect ABA biosynthesis under abiotic stress conditions [17]. These results may partially explain why, in this study, no significant differences in ABA content were observed in the OsASR6-RNAi plants (Figure 9). Moreover, at least two previous studies suggest that *ASRs*, serving as TFs, are able to directly bind specific *cis-elements* of the *NCED1* promoter, thereby affecting ABA biosynthesis and enabling adaptation to abiotic stress [13,33]. Interestingly, this study is the first to suggest interaction between an ASR and NCED1, and a subsequent effect on ABA biosynthesis in response to salt stress.

In summary, the results of this study collectively suggest that *OsASR6* plays a positive role in enhancing salt tolerance in rice. *OsASR6* was found to activate the antioxidant system and reestablish cellular ion balance in response to salt stress. Furthermore, OsASR6 was also found to regulate ABA biosynthesis by directly interacting with OsNCED1. Overall, these findings suggest that *OsASR6* is a candidate gene for genetic manipulation of new salinity-resistant lines in rice, highlighting a potential role in future breeding programs.

## 4. Materials and Methods

### 4.1. Plant Materials and Stress Treatments

Rice (*Oryza sativa japonica* cv. Zhonghua11) seedlings were grown in a greenhouse (Anhui Agricultural University, Hefei, China) under standard rice growing conditions. Fourteen-day-old rice seedlings were then treated with 100 mM ABA, 200 mM H_2_O_2_ or 150 mM NaCl. Root and shoot samples from 3-week-old rice plants were collected 0, 1, 3, 6 and 9 h after treatment, respectively, then stored at −80 °C for further analysis.

For salt stress treatment at the germination stage, wild-type (WT) and transgenic rice seeds were sterilized using 10% sodium hypochlorite then grown on ½ solid Murashige and Skoog (MS) (Yuanmu Biotech, Shanghai, China) medium enriched with 100 mM NaCl for 4 d. Germination rates were then determined. For post-germination salt stress treatment, 14-day-old WT and transgenic seedlings were transferred to Hoagland solution containing 130 mM NaCl for 14 d then standard Hoagland solution (Yuanmu Biotech, Shanghai, China) for a further 3 d. Survival rates, ion leakage, malondialdehyde (MDA) content, Na^+^/K^+^ contents and antioxidant enzyme activities were then determined 14 d after initiating salt stress. In addition, seeds were also germinated on 1/2 MS medium containing 2 µM methyl viologen (MV) for 6 d or 5 or 10 µM ABA for 14 d then shoot lengths were measured in order to analyze oxidative and ABA sensitivity.

### 4.2. Vector Construction and Production of Transgenic Rice Lines

Specific primers (5′-GG**GGTACC**ATGGCTGACGAGTACGGC**-**3′ and 5′-CG *GGATCC* TCAGCCGAAGAGGTGGTG-3′, *Kpn*I site bold, *Bam*HI site Italic) were used to amplify the full-length *ASR6* (LOC_Os04g34600.1) gene. Rice cDNA obtained from leaf samples served as the PCR template. The PCR products were then cloned into the pCAMBIA1390 vector under control of the Ubi promoter, and used to transform and produce transgenic overexpressing plants. The Ubi promoter was amplified using the following specific primers: (5′-GCC C**AA GCT T**CT AGT GCA GTG CAG CGT GAC-3′ and 5′-GCAA*CTGCAG*GTACCTAGTGCAGAAGTAACACCA-3′, *Hin*dIII site bold, *Pst*I site Italic), then inserted downstream of the 35S promoter.

Specific primers (5′-ATA**CTCGAG**TCGTGCTCAAAAGCCGGT-3′ and 5′-ATTAGATCTGAGGCGCTCCTTGTTCTT-3′, *Xho*I site bold, *Bgl*II site underlined) were also used to produce RNAi constructs. The PCR products (120 bp of *OsASR6* exon region) were then inserted upstream and downstream, respectively, in the PUC-RNA interference vector [34] in an inverted repeat configuration, generating inverted repeats containing an intron space. Confirmed units were then inserted into pCAMBIA 1300 to generate OsASR6-RNAi constructs. The constructs were then introduced into *Agrobacterium tumefaciens* strain GV3101. The transformation and the production of *OsASR6* transgenic rice plants were performed as described previously [35]. Two independent OsASR6-OE (OE19 and OE30) and OsASR6-RNAi lines (Ri5 and Ri11) possessing different transcription levels were used in the following analyses.

### 4.3. RNA Extraction, Reverse Transcriptase (RT-PCR) and Quantitative Real-Time PCR (qRT-PCR)

Total RNA was extracted using the RNAiso plus method (TaKaRa, Dalian, China). SuperScriptTM III reverse transcriptase was used for cDNA synthesis. RT-PCR was performed as follows: cDNA from WT and transgenic lines was used to amplify *ACTIN1* in order to screen the cDNA contents, which presented similar expression levels as *ACTIN1*. Identical amounts of cDNA were then used to amplify *OsASR6.* The PCR products were analyzed using agarose gel. qRT-PCR was conducted using an ABI PRISM 7500 sequence detection system according to the manufacturer’s manual. The 2^−ΔΔCt^ method was used to analyze the experimental data. Each set of RT-PCR and qRT-PCR was repeated four times. The primers 5′-GAAGGAGCACAAGAACAAGGAGC-3′ (forward) and 5′-GATGGATCATCAGCCGAAGAGGT′ (reverse), which are specific for OsASR6, were used for both RT-PCR and qRT-PCR. *OsACTIN1* was used as an internal control, and was amplified with the primers 5′-CTGACGGAGCGTGGTTACTCAT-3′ (forward) and 5′-TGGTCTTGGCAGTCTCCATTTC-3′ (reverse).

### 4.4. Subcellular Localization

Full-length cDNA from OsASR6 without the stop codons was inserted into a modified pCAMBIA1305 vector [36] in which the GUS region was replaced with GFP. pCAMBIA1305-GFP and pCAMBIA1305-OsASR6-GFP plasmids were then obtained, respectively. Specific primers 5′-GG**ACTAGT**ATGGCTGACGAGTACGGC-3′ and 5′-CG*GGATCC* TCAGCCGAAGAGGTGGTGC-3′ (*Spe*I site bold, *Bam*HI site Italic) were used to clone the *OsASR6* gene. The PCR products were then inserted into the pCAMBIA1305 vector using SpeI and BamHI enzymes, then the constructs were introduced into *Agrobacterium tumefaciens* strain GV3101. The resulting plasmids were transformed into tobacco or rice leaf protoplasts for analysis of subcellular localization [19,37]. After 24 h, observations of the transformed samples were then carried out under a Zeiss LSM 710 laser scanning confocal microscope (Jena, Germany) at 488 nm excitation.

### 4.5. Physiological Index Assays

Assays of relative electrolytic leakage (REL) and 3,3-diaminobenzidine (DAB) staining were performed based on Hu et al. [37], while measurements of Na^+^ and K^+^ were performed based on Fang et al. [29]. A Micro Hydrogen Peroxide (H_2_O_2_) Assay Kit (Beijing Solarbio Science & Technology Co., Ltd., Beijing, China) was used to detect contents of H_2_O_2_, while endogenous ABA contents were detected based on Liang et al. [19]. MDA, and superoxide dismutase (SOD), peroxidase (POD) and glutathione peroxidase (GPx) activity assays were carried out according to Park et al. [15] and Zhang et al. [38].

### 4.6. Transactivation Activity Assays

For the transactivation activity assay, the pGBKT7-OsASR6 fusion construct plus pGBKT7-53 + pGADT7-T as a positive control and pGBKT7-lam + pGADT7-T as a negative control were transformed into the yeast strain Y_2_HGold according to the manufacturer’s instructors (Clontech, Dalian, China). Yeast cells were then selected on SD/-Trp/-His/-Ade medium at 30 °C for a duration of 3 to 5 days.

### 4.7. Yeast Two-Hybrid and Bimolecular Fluorescence Complementation (BIFC) Assays

To screen proteins that interact with OsASR6, a pGBKT7-OsASR6 bait vector was constructed then transformed into Y_2_HGold yeast. Yeast cells containing the bait construct were then mated with samples from the rice cDNA yeast library (Clontech Laboratories, Palo Alto, CA). Mated cells were then streaked onto SD/-Ade/-His/-Leu/-Trp (QDO) plates for 3 to 7 days. Yeast clones growing on the selection medium were then selected for further analysis. To confirm the interaction between OsASR6 and OsNCED1, pGBKT7-OsASR6 and pGADT7-OsNCED1 plasmids were constructed and co-transformed into the AH109 yeast strain and selected on QDO medium. Primers for pGBKT7-OsASR6 were as follows: 5′-C**GAATTC**ATGGCTGACGAGTACGGCC-3′ and 5′-GC*GTCGAC*TCAGCCGAAGAGGTGGTGC-3′ (*Eco*RI site bold, *Sal*I site Italic), and for pGADT7-OsNCED1 were: 5′-C**GAATTC**ATGCAAAGGATTTGCCCTGC-3′ and 5′-CG*GGATCC* TCATTGGTGCTGTGACTGGAGC-3′ (*Eco*RI site bold, *Bam*HI site Italic).

To clarify the interaction, a BIFC assay was then carried out according to Chen [39]. Briefly, OsASR6-YFP^N^ and OsNCED1-YFP^C^ units were co-transformed into rice protoplasts then 24 h later, YFP signals were detected under a Zeiss confocal microscope (LSM 710, Jena).

### 4.8. Statistical Analysis

All data were analyzed for statistical significance using one-way analysis of variance (ANOVA) based on the Student’s *t*-test. Values were expressed as means ± standard deviation (SD) of three independent experiments

## Figures and Tables

**Figure 1 ijms-23-09340-f001:**
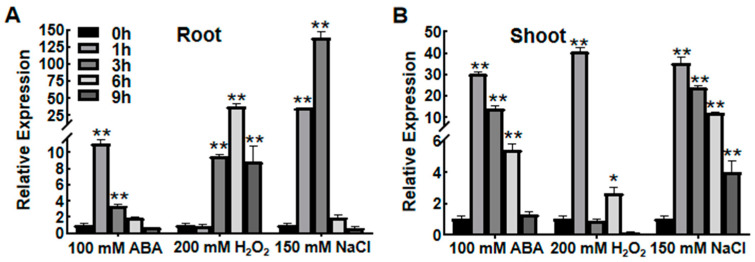
Expression profiles of *OsASR6*. Inducible expression pattern of *OsASR6* in (**A**) the roots and (**B**) the shoot under ABA, H_2_O_2_, and NaCl stresses treatment. Rice *Actin1* was used as an internal control for qRT-PCR analysis. Asterisks indicate significant difference compared to the transcription level of 0 h groups. Three independent experiments were performed. Data represent means ± SD. * *p* < 0.05, ** *p* < 0.01.

**Figure 2 ijms-23-09340-f002:**
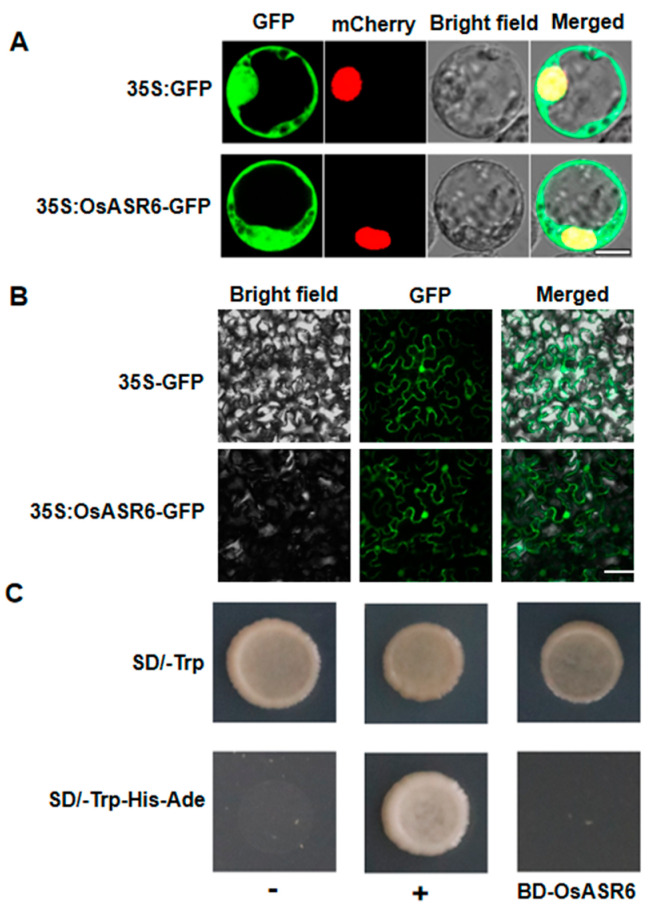
Subcellular localization and transcriptional self−activating activity of OsASR6. Localization of OsASR6 protein in (**A**) rice protoplasts (scale bar: 10 μm) and (**B**) tobacco leaf epidermal cells (scale bar: 20 μm). (**C**) Transactivation assay of OsASR6. +: pGBKT7-53/pGADT7-T (positive control), -: pGBKT7-lam/pGADT7-T (negative control).

**Figure 3 ijms-23-09340-f003:**
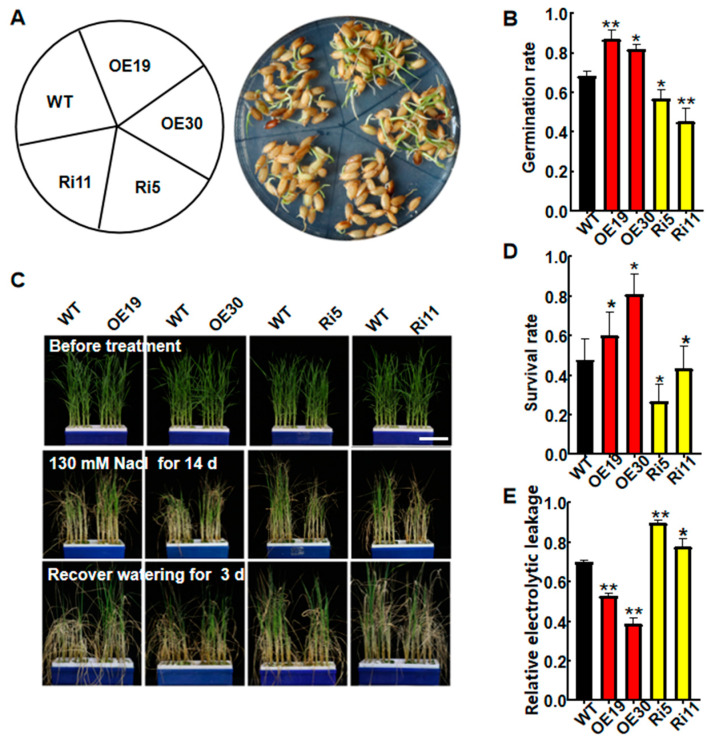
Analysis of salt tolerance in WT and *OsASR6* transgenic seedlings. (**A**) Phenotypes of wild-type (WT) and transgenic lines following 100 mM NaCl treatment for 4 d. (**B**) Germination rates of WT and *OsASR6* transgenic plants following 100 mM NaCl treatment for 4 d. Data represent means ± SD (*n* = 30). (**C**) Phenotypes of all tested rice plants before and after salt treatment, and after re-watering. Four-week-old seedlings were used for salt treatment (bar = 5 cm). (**D**) Survival rates and (**E**) relative ion leakage following NaCl treatment for 4 d. Asterisks indicate significant differences between means of WT and transgenic lines. Three independent experiments were carried out with similar results. Data represent means ± SD. * *p* < 0.05, ** *p* < 0.01.

**Figure 4 ijms-23-09340-f004:**
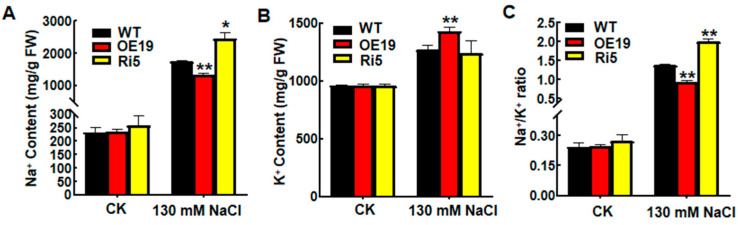
Analysis of Na^+^ and K^+^ content. (**A**) Na^+^ content, (**B**) K^+^ content, and (**C**) Na/K ratio of wild-type (WT) and *OsASR6* transgenic plants following 130 mM NaCl and control (CK) treatment, respectively. Three independent experiments were carried out with similar results. Asterisks indicate significant differences between means of WT and transgenic lines. Data represent means ± SD. * *p* < 0.05, ** *p* < 0.01.

**Figure 5 ijms-23-09340-f005:**
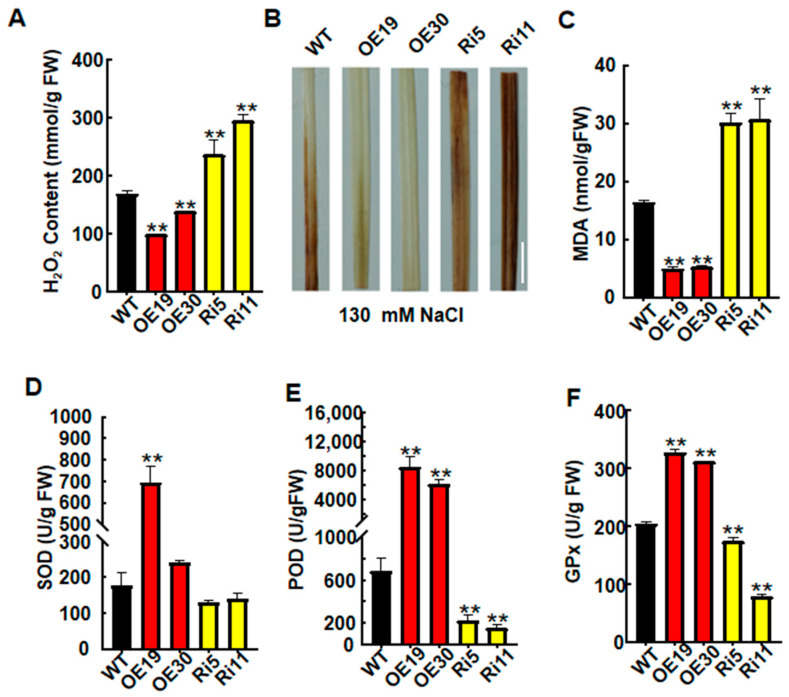
Analysis of physiological indexes in WT and *OsASR6* transgenic plants under salt treatment. (**A**) H_2_O_2_ content, (**B**) DAB staining, (bar = 1 cm), (**C**) the MDA content, and (**D**) SOD, (**E**) POD and (**F**) GPx enzyme activity. Asterisks indicate significant differences between means of WT and transgenic lines. Three independent experiments were carried out with similar results. Data represent means ± SD. ** *p* < 0.01.

**Figure 6 ijms-23-09340-f006:**
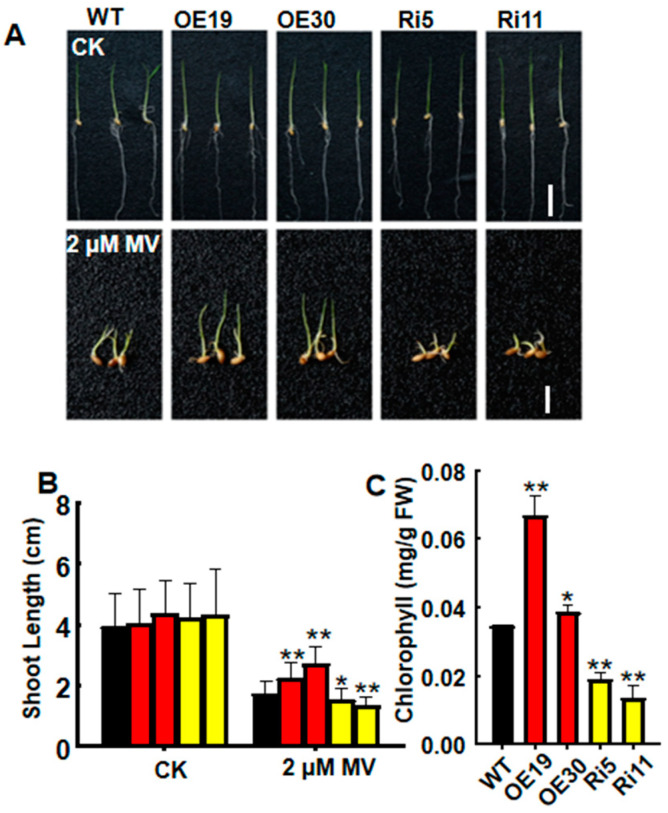
Effects of MV on WT and *OsASR6* transgenic rice. (**A**) Phenotypes of WT and transgenic seedling following treatment with 2 μM MV. Scale bar: 2 cm in the upper images, 1 cm in the lower images. (**B**) Shoot length and (**C**) chlorophyll contents following 2 μM MV and control (CK) treatment, respectively. Asterisks indicate significant differences between means of WT and transgenic lines. Three independent experiments were carried out with similar results. Data represent means ± SD (*n* = 30). * *p* < 0.05, ** *p* < 0.01.

**Figure 7 ijms-23-09340-f007:**
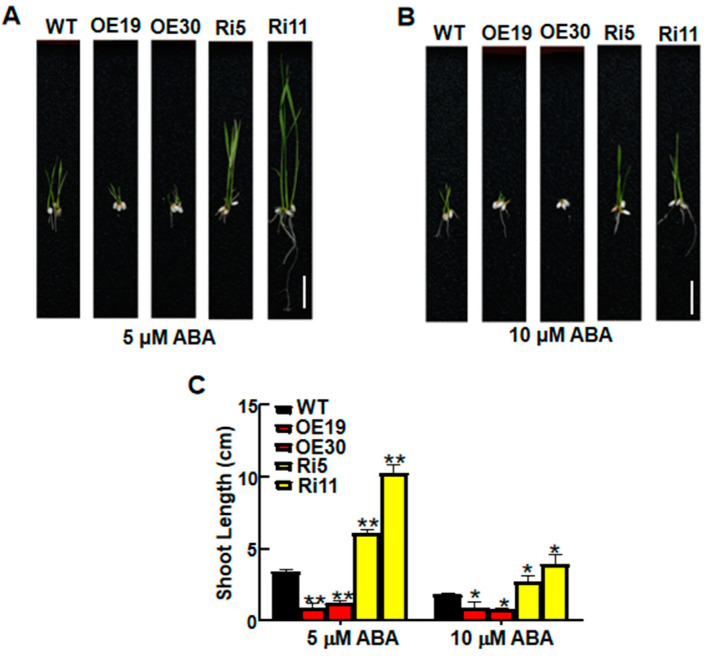
ABA sensitivity of WT and *OsASR6* transgenic rice. Phenotypes of wild-type (WT) and *OsASR6* transgenic seedlings following treatment with (**A**) 5 and (**B**) 10 μM ABA (scale bar: 3 cm). (**C**) Shoot lengths following each treatment. Asterisks indicate significant differences between means of WT and transgenic lines. Data represent means ± SD (*n* = 30). * *p* < 0.05, ** *p* < 0.01.

**Figure 8 ijms-23-09340-f008:**
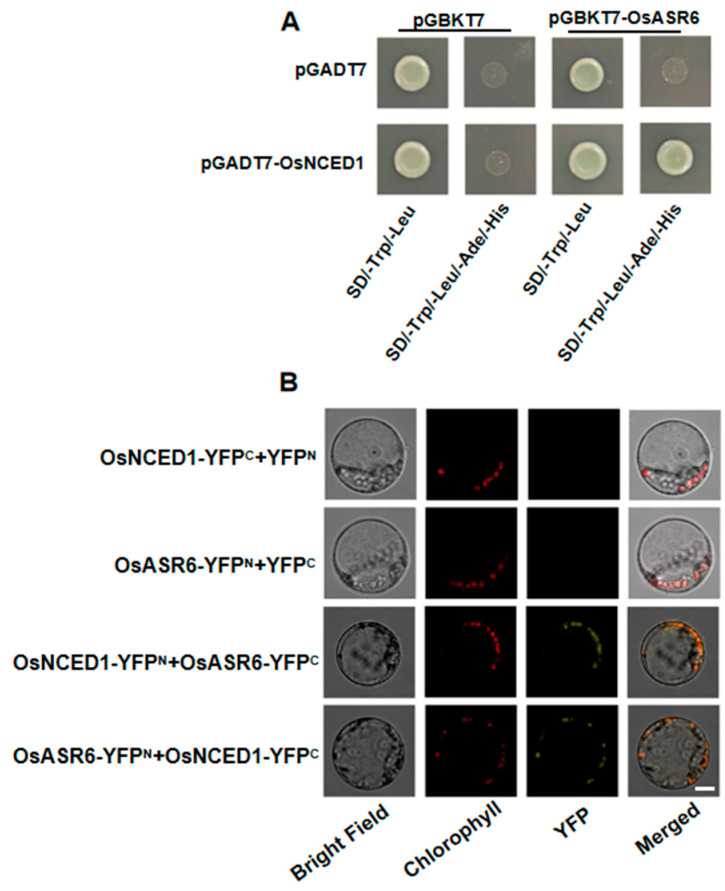
OsASR6 interacts with OsNCED1. (**A**) OsASR6 was found to interact with OsNCED1 in a yeast two-hybrid system. (**B**) A BIFC assay further confirmed the interaction in rice leaf protoplasts (scale bar: 10 μm).

**Figure 9 ijms-23-09340-f009:**
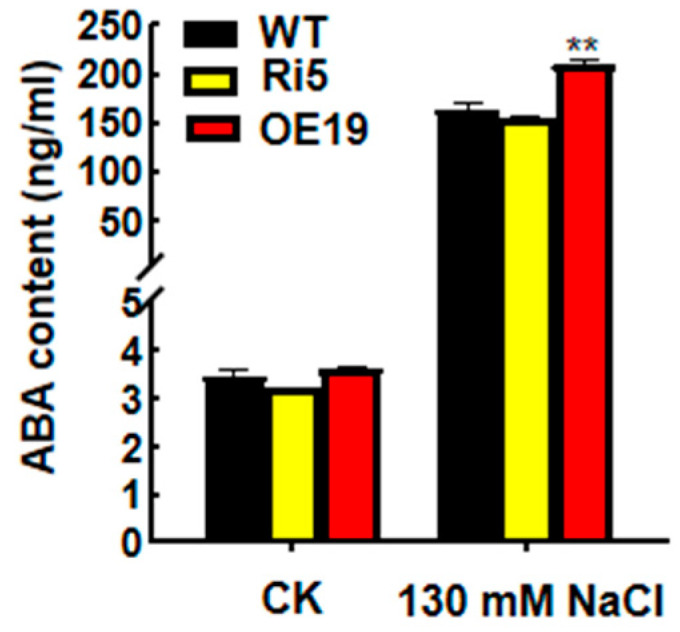
ABA contents in wild-type (WT) and *OsASR6* transgenic plants following treatment with 130 mM NaCl and control treatment, respectively. Three independent experiments were carried out with similar results. Asterisks indicate significant differences between means of WT and transgenic lines. Data represent means ± SD. ** *p* < 0.01.

## Data Availability

All data are displayed in the manuscript.

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
