# Peer review of "OsASR6 Enhances Salt Stress Tolerance in Rice"

_ijms, 2022, doi:10.3390/ijms23169340_

Round 1

Reviewer 1 Report

The paper by “OsASR6 enhances salt stress tolerance in rice” Qin Zhang, Yuqing Liu, Yingli Jiang, Aiqi Li, Beijiu Cheng and Jiandong Wu is devoted to investigation of the role of OsASR6 gene in salt stress resistance in rice plants.

The authors have carried out a thoughtful study, the results of the study are convincing and complete. The paper deserves the publication and represents a new knowledge about ASR genes family in higher plants.

However, the way of presenting the data seems to be rather negligent. Authors should significantly improve the text in order to be able to publish it. In general, the text needs numerous additions and clarifications to make the article understandable for the readers. Below is a list of notes on what needs to be corrected.

 1.      The introduction describes rather poorly the mechanism of participation of abscisic acid in signal transduction when the mechanisms of protection against salt stress are activated. This is one of the key points of your research. Please expand this part.

2.      In the section “Materials and methods” several methods used in the study are not described: RT-PCR, real-time RT-PCR with primers sequences, and Microscopy procedures according to Fig. 2. The description given in L.116 corresponds only to Fig. 8 and is not comprehensive.

3.      L. 91-93. Please describe the methods of determination the Survival rates, ion leakage, malondialdehyde content, Na+/K+ contents and antioxidant enzyme activities. Put references to the published methods or describe them in detail in this section indicating the formulas for calculating the relevant parameters.

4.      L.99-104. Please indicate what the bold and underlined nucleotides sequences in the primers’ sequences mean.

5.      L.131. It is unclear, what “SD/Trp/His/Ade medium” is. Besides, in L.137 the medium SD/-Ade/-His/-Leu/-Trp (QDO)” differently. Please, specify the composition of the medium and unify the spelling.

6.      There are abbreviations in the text which are not introduced at all. And besides, what is even more important, mentioning the corresponding proteins, methods, environments, etc. needs additional explanations. Otherwise, the mention of these abbreviations looks meaningless and incomprehensible to the reader.

-          L.53 – WDS domain. The mention of this domain requires the additional information concerning its function.

-          L.88 – MS medium

-          L. 123 – SOD, POD and GPx. In the section “Results”, which describes the measurements of these parameters, it is necessary to mention, why the corresponding measurements were taken and how these parameters are related to the determination of the level of oxidative stress.

-          L. 141 – BIFC assay. This method is not described in detail in the section “Materials and methods” and have no a reference. Either put a reference to the published method or describe it in detail in this section.

-          In Figures – Ri5, Ri11, OE19, OE30. Please, put in the section “Materials and methods” a definite designation of the mutants that you have used in the study and use the same names in the text and in the figures for easier to understanding your results

-          In Fig. 8 – AD and BK

7.      Some abbreviations in the text are introduced not at the first mention

-          L. 262  - OsNCED1 is introduced for the first time in the text as epoxycarotenoid dioxygenase 1. At that, in the end of Introduction, one of the main results of the study is: “In addition, OsASR6 was found to physically interact with Os-75 NCED1 both in vitro and in vivo.” The introduction should contain an information about this protein to make the reader understand the meaning of the result.

-          L.119 – “relative electrolytic leakage (REL)” and L. 221-222 - “3,3-diaminobenzidine (DAB)” should be in the L.119, where they are mentioned for the first time.

8.      L.158 – “These results suggest that OsASR6 is involved in multiple stress responses.” It is unclear, what you mean. Please, describe the conclusions from the result in a more understandable way. Don’t forget about the numerous cross-talks in plant signaling pathways.

9.      Please, pay attention to the numbering of sections in your article. All of them are marked with numbers 1 and 1.1.

10.  Please, change the titles of the legends in Figures 3, 5, 6 and 7 so that they would not be conclusions, but descriptions of the figures, as it is, for example, done in the legends for Figures 1, 2, 8, or include the words "effect of …"

11.  The conclusions from the results in Figures 6 and 7 need further clarification: what is the relationship between plant root length and abscisic acid metabolism. In addition, it is not clear, why methyl viologen was added to the growth media. What does it affect in terms of creating stress conditions and what is its relationship with the mechanisms of plant defense against salinity? Please add all these explanations to the text to improve the understanding of the results of the paper.

12.  Fig.S1, its legend and description in the text is very foggy and negligently done. For example, the differences between B and C and D and E are unclear. Please, make this figure, its legend and description in the text clear and understandable.

13.  The results of Fig. 9 (L.275-281) only show that the level of ABA increases in OsASR6 overexpressed mutants. These results have no any relation with NCED1. You are only able to discuss it in the section “Discussion” together with data on Fig. 8. Besides, it is missed in this part of “Results” the description of that there were no difference in ABA content in Ri5 and WT plants.

14.  The Discussion section should include not only the references to literature data, but also the references to Figures in this paper with relevant results. Please paste wherever you mention them.

15.  L. 300-301 “Collectively, these findings suggest that OsASR6 is a candidate gene of salt tolerance in rice”, as well as in the common summary of the study, L.355-356: “Overall, these findings suggest that OsASR6 is a candidate gene of salt tolerance in rice”. What did you mean? “OsASR6 is a gene, which may participate in ABA synthesis regulation in salinity stress tolerance in rice”? or “OsASR6 is a candidate gene for genetic manipulations for production of new salinity resistant lines of rice”?

16.  L. 305-309. The discussion of data concerning WRKY and MsPIP2;2 proteins without explanation of what are they and what are their functions makes this part of the text incomprehensible.

17.  L.334-335: “As a result, nuclear and cytoplasmic localization were confirmed, while transcriptional activation activity was absent, suggesting that OsASR6 is not a TF”.  Nowhere further in the text do you explain what it is in this case, if OsASR6 is one of the members of ASRs family, which are “thought to function as either molecular chaperones or TFs” (L.331). May be OsASR6 is a molecular chaperone for NCED1?

18.  L.335-337: “However, inter action between the NCED1and OsASR6 proteins was identified in chloroplasts of the rice protoplasts.” Please explain what data allow such a conclusion.

19.  L.337-338: “… OsASR5 with HSP40, OsASR5 and 2OG-Fe (II) in the protoplasts of rice and tobacco epidermal cells.” Is it so? The proteins from Oryza sativa was found in tobacco?

20.  L.344-346: “These results may partially explain why, in this study, no significant differences in ABA content were observed in the OsASR6-RNAi plants.” To make it clear what the authors mean, additional explanations are needed. Probably, the authors mean that this is not the only gene that affects the intensity of the synthesis of abscisic acid? And what happens in this case?

Author Response

Thank you for your letter and for the reviewers’ comments concerning our manuscript entitled, “OsASR6 enhances salt stress tolerance in rice”.  We found all comments be both valuable and helpful during the revision of our paper, as well as being of important significance in our ongoing research. We have studied the comments carefully and made corrections accordingly. Major corrections are highlighted in the revised manuscript and point-by-point responses to the reviewers’ comments are provided.

Reviewer 2 Report

I have completed the review of the manuscript entitled “OsASR6 enhances salt stress tolerance in rice.”  The topic of the study is very interesting, yet in my opinion there are many flaws in the research plan, presentation of data and analysis of results.  The detailed comments are given below:

·            In the methodology section, nothing is mentioned about the qRT-PCR analysis. It must be written completely. Moreover, how Na/K ratio was estimated is not mentioned in the manuscript.

·            Statistics missing in Fig 1. Gene expression data should also be analysed statistically using three biological replicates.

·            For the real time gene expression studies, only one reference gene has been used, however for accurate analysis minimum two reference genes should be used.

·            In Fig 3B: OE19 is significantly different from OE30 but not significantly different from Ri11 despite having lesser values that OE30. This seems to be wrong. Same issue is there with other figures as well. Thus, statistics needs to be revised.

·            Line number 158: In the results, authors have written that “these results suggest that OsASR6 is involved in multiple stress responses”, however only salinity stress treatments were given (ABA is a phytohormone and H2O2 is a ROS molecule, both are involved in growth and development also). How can authors give this statement without imposing any other stress treatment?

·            Line number 292: “In fact, transcription of OsASR6 (previously  named OsASR4) was found to be unaffected by drought treatment’, this statement needs more clarification. Please clarify whether the gene characterized in the present manuscript is OsASR4 or OsASR6.

·            Line number 192 : Why 130mM NaCl? any specific reason

·            In the present manuscript, only SOD, POD and GPX, ROS scavenging enzymes have been analysed, however analysis of other enzymes especially APX and CAT is necessary as H2O2 has more affinity for APX. So, these enzymes should also be analysed.

·            Serial number of subtitles is not correct, kindly reconcile your section numbering.

·            There are number of spelling mistakes throughout the text. Language needs improvement.  There are number of grammatical mistakes as well.

·            This manuscript must be reviewed by co authors or colleagues and by an expert, native English-speaking scientist.

·            The abstract needs to be revised and improved

·            In line number 30, delete ‘comma’ after  ‘water [2]’.

·            In line number 40, replace ‘helps regulate’ with ‘helps in regulating’

·            In line number 42, replace ‘stress’ with ‘stresses’

·            In line number 94, rationale of using methyl viologen (MV) is missing from the manuscript.

·            In line number 145: Spelling of ‘analysis’ is wrong.

Author Response

(The authors gave the same response as above.)

Reviewer 3 Report

Manuscript entitled “OsASR6 enhances salt stress tolerance in rice” by Zhang et al. address the study of the Abscisic acid-, Stress-, and Ripening-induced (ASR) proteins in rice. In this manuscript the authors have identified a new salt-induced ASR gene in rice (OsASR6) and functionally characterized its role in mediating salt tolerance. The authors approach the study of this gene through different approaches, consisting of the creation of overexpressing and silenced plants, and the observation in them of different morphological, physiological and stress-related parameters. The authors test different treatments (ABA, H2O2 or NaCl) and interestingly, this study is the first to suggest interaction between an ASR and NCED1. The results obtained are interesting, but some issues need to be revised throughout the manuscript. The experimental design and the proposed approaches used in this work seem both correct for the most part. However, in my opinion, the manuscript presents some issues mainly in the Materials and Methods, and other things, that need to be improved to make this work suitable for publication.

1. Main issues:

1.1. Regarding Introduction:

Line 69. Rice has 6 ASR genes and since the nomenclature is different depending on the articles or websites that are consulted, it should be specified which ASR gene is in this work (ASR1, Os01g72900; ASR2, xxxxxxxxxx). The introduction indicates that the ASR6 gene will be used, and the discussion indicates that it was previously ASR4. The authors should have to correct the nomenclature accordingly.

1.2. Regarding Material and Methods:

I must point to the authors that, according to the Instructions for Authors section of this journal, Full experimental details must be provided so that the results can be reproduced.  The Materials and Methods section of this manuscript is not detailed enough for other authors to repeat some of the experiments, in particular the molecular experiments (cloning, RT-PCRs, etc.). See my additional comments above.

1.3. Regarding Discussion:

Line 336. if OsASR6-GFP according to Figure 2 is nuclear and cytosolic, why now in Figure 8 does it seem to be plastidial? The authors should address this question and try to suggest a plausible explanation.

2. Additional comments:

2.1. Regarding Material and Methods:

Line 80. Please correct “1. Materials and methods” with “2. Materials and methods”

Please correct the numeration of the other sections accordingly (lines 81, 98, 111, 118, 126, 133, 145, 150, 151, 164, 183, 215, 247, 287)”

Line 85. I have doubts about the concentration of NaCl used in some experiments (100, 130 or 150). The authors should specify which concentration are used in each case.

Lines 88-96. MS medium should be detailed or referenced. Is the medium solid? Is the medium sterile? If the germination experiments are carried out under in vitro conditions, what is the seed sterilization protocol? Also Hoagland's solution should be detailed or referenced. Are the seedlings used in the post-germination salt stress treatment previously acclimated?

Lines 99 to 109. This section is an example of the lack of important information to reproduce the experiments described. What kind of DNA template is put into the PCR reactions? Specify restriction enzymes.  Specify Agrobacterium strain. Specify selection agent.  Specify sizes and composition of amplified regions. Is it cloned from the promoter to the terminator? Which regions are amplified for RNAi? Please, fill the appropriate details in this case.

Line 112. Replace “deleted stop codes” with “without the stop codon”

Line 113. pCAMBIAI1305 is misspelled. The plasmid pCAMBIA1305 carries GUS and not GFP. What type of pCAMBIA1305 do you use? The authors have used a modified pCAMBIA1305? In this case this modification should be detailed or referenced.

Lanes 112-117. Specify primers and cloning strategy. Specify DNA template, specify Agrobacterium strains. Specify methods to obtain protoplasts and transformation.

Line 119. I think that the REL staining method is not described in the provided reference

Lines 126-132. The cloning procedure to obtain the different constructions should be detailed.

Lines 134-143. The origin of the yeast library should be referenced or described. Again, details about the cloning procedure perform to obtain the different constructions is missing.

2.2. Regarding Results:

Line 156. The description of OsASR6 induction after H2O2 treatment is not accurate. In this case no increase is seen after 3 h  of H2O2 treatment.

Figure 1 legend. Nothing is specified in the Materials and Methods section about how these experiments are done, how plants are grown, age of plants, primers, RNA extraction, type of analysis to calculate relative expression. n = ?

Line 166. The 35S::OsASR6:GFP vector is the pCAMBIA1305-OsASR6-GFP plasmid? I kindly urge to the authors to standardize the nomenclature of the constructs use.

 Line 179. OsASR6 protein or OsASR6-GFP protein?

Line 186. qRT-PCR and RT-PCR experiments are not detailed in the Materials and Methods section.

Line 186. What parameter has been used to measure germination rate?

Line 199. Please correct “Fig. 4A & C” with “Fig.4 A-C”

Figure 3C. WT plants should behave in a similar way in the different experiments showed in this figure. The WT plants in the experiment of 130 mM NaCl show a much worse phenotype in the case of the experiment with overexpressing plants as compared to the experiment with Ri plants. The same situation appears in the experiment with recover watering. Looking these variations in WT plants I have doubts about the results obtained.

Line 209. Figure 3 legend. The number of replicates (n) is missing

Line 213. Figure 4 legend. The number of replicates (n) is missing

Figure 5B. Size bar is missing.

Line 240. Figure 5 legend. The number of replicates (n) is missing

Line 245. Figure 6 legend. The “Data represent means  ± SD (n=?)” is missing ".

Figure 7A-B. to maintain the homogeneity in the figures, in panels A and B, OE19 and OE39 should appear before Ri5 and Ri11. Please modify.

Line 284. Figure 9 legend. The “Data represent means  ± SD (n=?). **P…..” is missing ".

2.3. Regarding Discussion:

Line 326. In the sentence there is an extra space between the words “salt due”

2.4 Regarding References:

The references list have noumerous format mistakes. In fact, none of the 36 references are in the journal format. There is a space left between the initials of the names of the authors. Also, there is an extra comma after the last initial of the last author, that should be remove. The volume should be italicized. The issue of the journal should not appear behind the volume (this sometimes occurs; e.g. reference 5). The hyphen separating the pages should be long (see recent publications of the journal). Many titles include names of organisms or genes that should be italicized (e.g. Gossypium hirsutum in reference 2, Panicum virgatum in reference 5, Os2H16 or cis in reference 18, etc.).

2.5. Regarding Supplemental Figure 1:

The schematic representation of the RNAi and overexpression constucts are bad. In the plasmid pCAMBIA1390 there is no Ubi promoter. Also, in this vector HPTII gene is located near to the LB border. In the two  schematic representations of the constructs the direction of the 35s promoter does not control expression of any gene.  What does “inton” mean in the RNAi representation. In panel C, check “Ri11”

Line 360. Figure S1 legend. Please correct “…of the RNAi and overexpression constructs”  with  “…of the RNAi (up) and overexpression (down) constructs”

Author Response

(The authors gave the same response as above.)

Round 2

Reviewer 1 Report

The authors of the present study made the most corrections, but a number of corrections were not done. The authors of the paper should overcome the misconception that any reader has the same knowledge as they do, and that everyone is easily able to guess what they mean by abbreviations and why this or that experiment was required. All necessary explanations should be given in the text to facilitate its understanding for any reader.

1.      Some abbreviations in the text are still not introduced at all

– SOD, POD and GPx

– BIFC

2.      OsNCED1 is now introduced at the first mention in the text as epoxycarotenoid dioxygenase 1. However, the conclusion in the end of Introduction is still not clear, since it is not written that OsNCED1 is a key enzyme of ABA biosynthesis.

3.      The part of Results, which concerns the addition of methyl viologen to the growth media, is still unclear for a reader, since it is not obvious that MV transfers electrons from the photosynthetic electron transport chain to oxygen, thus producing the formation of reactive oxygen species and thus creating oxidative stress conditions in plants. These explanations must be written in Results.

4.      The legend to Fig. 1S is still foggy and negligently done. I cannot understand the differences between B and C, D and E.

5. Besides, D and E are obviously not the Expression levels! These are the results of electrophoresis in agarose gels of PCR products obtained after RT-PCR. The description of electrophoresis in agarose gels of PCR products in the section of Methods is also lost.

6.  In the previous review, I have pointed that “The Discussion section should include not only the references to literature data, but also the references to Figures in this paper with relevant results. Please paste wherever you mention them.” That was not done at all.

-     L.334-335. It should be: “Transcription of OsASR6 was strongly induced by ABA, H2O2 and high salinity, suggesting a potential role in abiotic stress resistance (Fig. 1).”

-     L.335-338. It should be: “Consistent with this, OsASR6 overexpressing lines showed improved germination and survival rates under high salinity treatment, in contrast to the OsASR6-RNAi lines, which displayed opposite effects (Fig. 3A,C).”

-L.350-353. “Similarly, in this study, the OsASR6 overexpressing lines presented a reduced ratio of Na/K through reductions in Na+ and increases in K+ under salt stress conditions, while the OsASR6-RNAi plants showed a increased Na/K ratio [24, 32, 33]”.  These results are in Fig.4 of the present paper. The authors also indicate “in this study” However, they refer to other works. That is unclear. It seems like it should be “…showed an increased Na/K ratio (Fig. 4), as it has been previously shown in [24, 32, 33]”. Please write correctly.

- L.361-363. It should be: “Compared with the WT, OsASR6-overexpressing plants presented a lower MDA content and REL rates, and higher activities of SOD, POD and GPx under NaCl treatment (Fig. 5)…”

-L.375-376. It should be: “As a result, nuclear and cytoplasmic localization were confirmed (Fig. 2A,B), while transcriptional activation activity was absent (Fig. 2C), suggesting that OsASR6 is not a TF.”

-L.386-388. It should be: “These results may partially explain why, in this study, no significant differences in ABA content were observed in the OsASR6-RNAi plants (Fig. 7C).

Author Response

Thank you for your comments concerning our manuscript entitled, “OsASR6 enhances salt stress tolerance in rice”.  We found all comments be both valuable and helpful during the revision of our paper, as well as being of important significance in our ongoing research. We have studied the comments carefully and made corrections accordingly. Major corrections are highlighted in the revised manuscript and point-by-point responses to the reviewers’ comments are provided.

Reviewer 2 Report

Authors have addressed some of the comments but still there are mistakes especially in the statistics. Authors are advised to take help from a statistics expert. Besides, if APX and other enzymes were analysed, give that data in the manuscript ( in supplementary data).  There are number of spelling mistakes throughout the text. Language needs improvement.  There are number of grammatical mistakes as well.

Author Response

(The authors gave the same response as above.)

Reviewer 3 Report

The authors have made a considerable improvement in the manuscript but there are still things to modify. In my opinion, some critical issues remain unaddressed, mainly regarding the description of the cloning strategies performed, and other things, that need to be improved to make this work suitable for publication.

1. Main issues:

1.2. Regarding Material and Methods:

Line 107. The strategy of insertion of the Ubi promoter into the pCAMBIA 1390 vector should be detailed or referenced.

Lines 109 to 114. PUC-RNA interference vector should be referenced. The cloning of the inverted repeat in pCAMBIA 1300 to generate OSASR6-RNAi constructs should be detailed.

Lines 121 to 130. Although the experimental details of qRT-PCR experiments have been detailed in the revised version, it does not happen the same with the RT-PCR experiments. I must remind the authors that in the material and methods section should include all the necessary details to reproduce their experiments and this cannot be said yet for the RT-PCR assays carried out.

Line 134 to 137. The elimination of the GUS and the cloning of the GFP region should be detailed or referenced. The cloning procedure to insert the OsASR6 cDNA should be detailed because the primers have not restriction sites. If the SpeI and BamHI cutting sites are used, the 35S promoter is eliminated from the pCAMBIA 1305 vector. So…which promoter will direct the expression of OsASR6-GFP?

Lines 154 to 172. Please, specify primers and cloning strategy for the different genes (ASR6 and NCED1) and plasmids (pGBKT7 and PGADT7).

2. Additional comments:

2.1. Regarding Material and Methods:

Line 127. Please correct “study.Each”  with  “study. Each”

In different lines and Figure S1. Correct the nomenclature of restriction enzymes with the initials of the species in italics (for example: please correct “KpnI or BamHI”  with  KpnI or BamHI”)

Author Response

(The authors gave the same response as above.)

Round 3

Reviewer 1 Report

The authors have corrected the MS according to all of my suggestions. 

L. 161 Please, change glutathionepcmxidase to Glutathione peroxidase

Author Response

Thank you for  the reviewers’comments concerning our manuscript entitled, “OsASR6 enhances salt stress tolerance in rice”. We have studied the comments carefully and made corrections accordingly. Minor corrections are highlighted with blue in the revised manuscript and point-by-point responses to the reviewers’comments are provided.

Reviewer 2 Report

I appreciate the authors for revising the manuscript as per suggestions, however I am personally not convinced with the statistics yet. Like in Fig 1A, how comparisons were made as 1 hr and 3 hr samples reading are significantly different and still they are shown as non-significant. Similar problems are there with Fig 1B also. Moreover, I suggest the authors to mention it in the fig legend that comparisons were made only between wild versus OE and wild versus rnai lines.

Author Response

(The authors gave the same response as above.)

Reviewer 3 Report

The authors have made a considerable improvement in the manuscript. In my opinion it is ready to be accepted

Author Response

(The authors gave the same response as above.)
